# PS-SiZer map to investigate significant features of body-weight profile changes in HIV infected patients in the IeDEA Collaboration

Jaroslaw Harezlak[1], Samiha Sarwat[2], Kara Wools-Kaloustian[3], Michael Schomaker[4], Eric Balestre[5], Matthew Law[6], Sasisopin Kiertiburanakul[7], Matthew Fox[8], Diana Huis in 't Veld[9], Beverly Sue Musick[10], Constantin Theodore Yiannoutsos[11]*

1 Department of Epidemiology and Biostatistics, Indiana University School of Public Health, Bloomington, IN, United States of America, 2 Bayer U.S., LLC, Whippany, NJ, United States of America, 3 Department of Medicine, Indiana University School of Medicine, Indianapolis, IN, United States of America, 4 Centre for Infectious Disease Epidemiology and Research, University of Cape Town, Cape Town, South Africa, 5 Inserm, Institut de Santé Publique d'Epidemiologie et de Développement, Bordeaux, France, 6 Biostatistics and Databases Program, Kirby Institute, University of New South Wales, Sydney, Australia, 7 Department of Medicine, Faculty of Medicine, Ramathibodi Hospital, Mahidol University, Bangkok, Thailand, 8 Departments of Global Health and Epidemiology, Boston University School of Public Health, Boston, MA, United States of America, 9 Department of Internal Medicine and Infectious Diseases, University Hospital, Ghent, Belgium, 10 Department of Biostatistics, Indiana University School of Medicine, Indianapolis, IN, United States of America, 11 Department of Biostatistics, Indiana University Fairbanks School of Public Health, Indianapolis, IN, United States of America

* cyiannou@iu.edu

**Data Availability Statement:** Regarding data sharing, complete data for this study cannot be publicly shared because of legal and ethical

## Abstract

### Objectives

We extend the method of Significant Zero Crossings of Derivatives (SiZer) to address within-subject correlations of repeatedly collected longitudinal biomarker data and the computational aspects of the methodology when analyzing massive biomarker databases. SiZer is a powerful visualization tool for exploring structures in curves by mapping areas where the first derivative is increasing, decreasing or does not change (plateau) thus exploring changes and normalization of biomarkers in the presence of therapy.

### Methods

We propose a penalized spline SiZer (PS-SiZer) which can be expressed as a linear mixed model of the longitudinal biomarker process to account for irregularly collected data and within-subject correlations. Through simulations we show how sensitive PS-SiZer is in detecting existing features in longitudinal data versus existing versions of SiZer. In a real-world data analysis PS-SiZer maps are used to map areas where the first derivative of weight change after antiretroviral therapy (ART) start is significantly increasing, decreasing or does not change, thus exploring the durability of weight increase after the start of therapy. We use weight data repeatedly collected from persons living with HIV initiating ART in five regions in the International Epidemiologic Databases to Evaluate AIDS (IeDEA) worldwide collaboration and compare the durability of weight gain between ART regimens containing and not containing the drug stavudine (d4T), which has been associated with shorter durability of weight gain.

restrictions. The principles of collaboration under which IeDEA multi-national collaboration was founded and the regulatory requirements of the different countries' IRBs and other legislative and regulatory bodies, require the submission and approval of a project concept sheet by investigators, both within and outside of IeDEA, which has to be approved by the individual regions and the IeDEA Executive Committee as well as the principal investigators at the individual sites. For more information and helpful resources, please see https://www.iedea.org/resources/administrative-resources/ where a number of documents aiding the submission of multi-regional concept proposals can be requested. Proposals to individual regions are governed by similar processes (see for example https://www.ccasanet.org/collaborate/ for helpful documents and processes governing the Central, South America and the Caribbean Network, one of the seven IeDEA regions as well the concept proposal form for the East Africa IeDEA region, where the corresponding author's home region. The accuracy of the data are governed by each region's Regional Data Center, the IeDEA Executive Committee and the Data and Harmonization Working Group within IeDEA.

**Funding:** National Institute of Allergy and Infectious Diseases AI069911 Dr. Kara Wools-Kaloustian, Indianapolis, IN, USA National Institute of Allergy and Infectious Diseases AI069924 Dr. Matthias Egger, Bern Switzerland National Institute of Allergy and Infectious Diseases AI069907 Matthew Law, Foundation of AIDS Research, NY, USA National Institute of Allergy and Infectious Diseases AI069927 Tyler Hartwell, Research Triangle Park, NC, USA National Institute of Allergy and Infectious Diseases AI069923 Catherine McGowan, Nashville, TN, USA National Institute of Allergy and Infectious Diseases AI069919 Francois Dabis, Bordeaux, France.

**Competing interests:** The authors have declared that no competing interests exist.

## Results

Through simulations we show that the PS-SiZer is more accurate in detecting relevant features in longitudinal data than existing SiZer variants such as the local linear smoother (LL) SiZer and the SiZer with smoothing splines (SS-SiZer). In the illustration we include data from 185,010 persons living with HIV who started ART with a d4T (53.1%) versus non-d4T (46.9%) containing regimen. The largest difference in durability of weight gain identified by the SiZer maps was observed in Southern Africa where weight gain in patients treated with d4T-containing regimens lasted 59.9 weeks compared to 133.8 weeks for those with non-d4T-containing regimens. In the other regions, persons receiving d4T-containing regimens experienced weight gains lasting 38–62 weeks versus 55–93 weeks in those receiving non-d4T-based regimens.

## Discussion

PS-SiZer, a SiZer variant, can handle irregularly collected longitudinal data and within-subject correlations and is sensitive in detecting even subtle features in biomarker curves.

## Introduction

In the study of changes in longitudinal biomarkers in response to therapy or disease progression, it is useful to be able to identify the periods in time where changes occur. A key challenge arising from this effort is to isolate the underlying features of interest (say marker decreases or increases) in the presence of potentially large data variation. For example, in a data set of weight measurements in HIV-infected individuals initiating antiretroviral therapy (ART), which forms the core illustration of our methods in this paper, a scatterplot involving a mere 1% of the data (Fig 1 top left panel) is largely indecipherable. The situation does not improve when a "spaghetti" plot is generated (Fig 1, top right panel). However, a plot of the median weight at binned time points (Fig 1, bottom left panel) starts picking up the rapid early weight gains following ART initiation, but is less informative about the time when these gains reach a plateau and the possibility of long-term weight decreases possibly resulting from treatment toxicity or treatment failure.

The bottom right panel in Fig 1 includes three smooth weight trajectories at different values of a smoothing parameter estimated via a penalized spline regression method (see Ruppert et al., [1]), which appear to capture the well-known features in such data involving rapid weight increase and subsequent plateau [2]. However, it is unclear what the durability of weight gain is or whether there are decreases in weight after long-term exposure to therapy. In addition, each smoothing level produces a slightly different fit, particularly with respect to the timing of reaching the plateau in weight gain. As noted in Marron and Zhang [3], a hurdle in the application of smoothing methods is the selection of the smoothing parameter, because interesting features that are present in the data may be visible after applying some smoothing techniques or at some levels of smoothing but disappear in others, so choosing among the various smoothing techniques or the level of smoothing can be critical in extracting relevant features from the data; and of course, there is a tremendous computational burden associated with such data analyses, as the above conclusions were drawn from only about 1% of the underlying database.

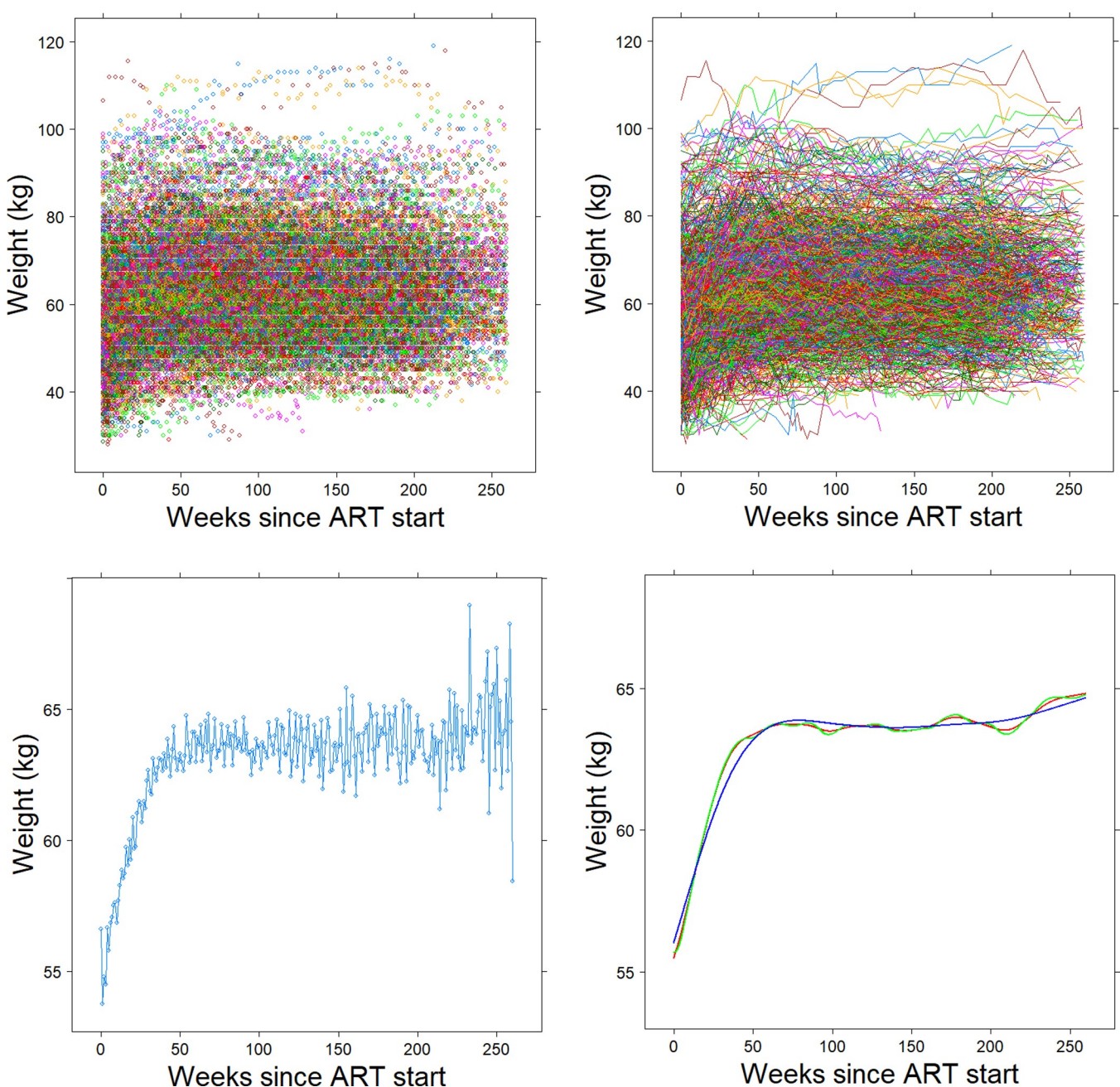

**Fig 1. Four different visualizations of weight changes t (kg) after antiretroviral therapy initiation in involving data from 1% of HIV-infected patients from the IeDEA database (2,000 patients, 46,207 observations).** The upper-left panel (a) represents a scatter plot; the upper-right panel (b) shows a spaghetti plot; the lower-left panel includes a plot of the mean weight over time; Lower-right panel represents smooth curves estimated at 3 different smoothing parameter values.

The Significant Zero Crossings of Derivatives (SiZer) [4] was proposed to address many of the aforementioned issues [5]. It is a useful Exploratory Data Analysis tool for understanding the significant features resulting from smoothed curves. SiZer simultaneously studies a family of smooth curves under a wide range of smoothing parameters (bandwidths) and produces inference on a smoothed version of the underlying curve viewed at varying levels of smoothing. The standard implementation of SiZer [5] is based on the local linear smoother with a

kernel-type smoothing method for a single predictor and a single outcome data [6]. The SiZer map graphically explores structures in curves under study by mapping areas where the curve is significantly increasing, decreasing or does not change by studying its first derivative. Statistical inference is based on the derivatives of the smoothed curve by constructing confidence intervals at each location and also at each level of the smoothing parameters. The technique assembles these analyses at a wide range of smoothing parameters and synthesizes them in a single "map" where increase, decrease and plateau regions are identified by different colors, presenting an attractive global visualization of the data under many smoothing scenarios.

A number of extensions of the SiZer methodology have been proposed to increase inference precision [7, 8]. Of greater relevance to this paper is the extension proposed by Park and colleagues [9], which relaxes the assumption of independent errors, thus ignoring spurious features which are caused by the existence of dependence in the data. These and other authors further extended the SiZer method into the area of time series [10, 11]. Another relevant extension is the SiZer for Smoothing Splines (SS-SiZer) [3], which uses splines to enhance detection of true features in the data.

Despite its attractiveness as a data visualization tool, the SiZer map has not been used widely in biological applications. This is unfortunate, since biological processes frequently involve changes in various measures (most notably biomarkers), which evolve over time, in response to disease progression or initiation and/or modification of clinical therapies. One important reason for this is the fact that the SiZer and its extensions do not account for within-subject correlation. This frequently arises in longitudinal settings from measurements obtained repeatedly on the same individuals. A further technical complication is that the timing of these measurements becomes increasingly less regular with the passing of time (as subjects miss or reschedule clinical visits). It should be clear that this is a much different problem from time-series analysis, since longitudinal data are obtained from the same sample of study subjects repeatedly over time. Thus, neither the originally proposed method of SiZer nor its extensions in the area of time series fully address the challenges posed by longitudinal data.

It is a core aim of this paper to extend the method of SiZer maps to account for within-subject correlation in the setting of irregularly collected longitudinal biomarker data, since SiZer offers an appealing global data visualization technique which can be tremendously useful in answering many important biological questions about the evolution of these data over time. We accomplish this by proposing a semiparametric extension of the SiZer methodology, named Penalized Spline SiZer (PS-SiZer), which combines a penalized spline regression model [10] with an embedded linear mixed-model representation of the marker evolution, coupled with methods which increase the computational efficiency of the standard SiZer. These computational advances are particularly attractive when analyzing massive databases with hundreds of thousands of patients and millions of observations.

The paper is organized as follows: In the Methods we give a brief overview of the core ideas of the SiZer methodology as originally proposed [1, 5] along with the SS-SiZer methodology of Marron and Zhang [3] and present the proposed penalized spline PS-SiZer procedure for longitudinal data. Two simulation studies are presented in the Results where the proposed methodology is compared with the local linear smoother LL-SiZEr [1, 5] and the SS-SiZer [3] in, respectively, detecting changes and plateaus in longitudinal biomarker data. These are followed by the analysis of a large database obtained from hundreds of thousands of HIV-infected patients enrolled in care and treatment programs around the world participating in the International Epidemiology Databases to Evaluate AIDS (IeDEA) Collaboration, where body weight measurements were collected repeatedly at each clinic visit. The clinical interest of this analysis is to describe the pattern of body weight changes after initiation of antiretroviral therapy as a surrogate of treatment effectiveness and to determine the durability of weight gain by

detecting a plateau in weight increases and the presence of possible decreases after long-term exposure to various therapeutic modalities. We conclude the paper with a brief discussion of our findings.

## Methods

### SiZer

More formally, for a given set of observed data $\{(X_i, Y_i)_{i=1}^n\}$ and a smoothing function $g(x)$, we can consider a non-parametric regression model as follows:

$$y_i = g(x_i) + \epsilon_i, \ i = 1, \ldots n, \ \epsilon_i \sim N(0, \sigma_\varepsilon^2) \tag{1}$$

Here, $g(x)$ is some "smooth" regression function that needs to be estimated from the data and $\epsilon_i$ is the random error component with variance $\sigma_\varepsilon^2$. The smooth function $g(x)$ may be a non-parametric regression function indexed by a smoothing parameter $\lambda$ (bandwidth) as $g_\lambda(x)$ [1].

### Local linear smoother SiZer: LL-SiZer

The LL-SiZer model specification [5] considers a family of smooth functions indexed by the smoothing parameter $\lambda : \{\hat{g}_\lambda(x) : \lambda \in [\lambda_{min}, \lambda_{max}]\}$ as in Ruppert, Wand and Carroll [1]. A significant feature in the data is detected from the confidence limits of the first derivatives of the fitted model $\hat{g}_\lambda$ at each level of $\lambda$.

The LL-SiZer applies the local linear regression method of Fan and Gijbels [12] to estimate $g_\lambda(x)$ and its derivative, $g'_\lambda(x)$. A common estimate of $g_\lambda(x)$ at each location of $x$ is given by the equation

$$\hat{g}_\lambda(x) = argmin \sum_{i=1}^n [y_i - \{a_0 + a_1(x_i - x)\}]^2 \times K_\lambda(x - x_i)$$

where *argmin* is the minimum of the sum jointly over the regression coefficients, $a_0$ and $a_1$. A line is fitted to the data for each $x$ using $K_\lambda$-weighted least-squares, where $K(\cdot)$ is a Gaussian kernel. A SiZer map is then constructed by changing the value of the smoothing parameter $\lambda$. The estimated regression function of $g_\lambda(x)$, and $g'_\lambda(x)$ are obtained to construct a family of smooth functions at various levels of the smoothing parameter. Confidence limits for $g'\lambda(x)$ are obtained as

$$\hat{g}'_\lambda(x) \pm q_\lambda \times \hat{sd}(\hat{g}'_\lambda(x))\}$$

where $q_\lambda$ is a suitably defined Gaussian quantile [7]. In the SiZer map, a pixel at $x$ and a specific smoothing level $\lambda$ is colored blue if the confidence interval suggests that $g'^{(x)} > 0$ (implying that the curve at $x$ is increasing, red if the confidence interval suggests that $\hat{g}'_\lambda(x) < 0$ (implying that the underlying curve is decreasing) and purple if the confidence interval contains zero (implying that no significant change in the curve can be detected).

### SiZer for smoothing splines (SS-SiZer)

The SiZer for Smoothing Splines (SS-SiZer) [3] is an extension of a kernel-type estimation to the smoothing spline estimation. SS-SiZer incorporates the smoothing spline model and estimates the regression function by minimizing

$$[y_i - g_\lambda(x_i)]^2 + \lambda \int [g''_\lambda(x)]^2 dx$$

where $\lambda$ is the smoothing spline parameter that determines the smoothness of the regression estimate $\hat{g}_\lambda(x)$ and $\int \left[ g_\lambda''(x) \right]^2 dx$ represents the roughness of the underlying function $g_\lambda(x)$. Here, the smoothing spline function $g_\lambda(x)$ is a natural cubic spline with knots at data locations $x_1 \ldots x_n$. The smoothing parameter, $\lambda$ acts similarly as the bandwidth in the LL-SiZer presented in the previous section.

SS-SiZer constructs point-wise confidence limits to produce the map. In our research, we apply the simultaneous confidence limit to the SS-SiZer model to address the multiplicity comparison issue (see next section). Otherwise, the interpretation remains the same as in the SS-SiZer maps [3]. For other implementation details, such as the expression of first derivative estimate and its standard error, the reader is referred to the paper by Marron and Zhang [3].

## The penalized SiZer (PS-SiZer)

In this section, we present our extension of the SiZer map to handle data that arise in the longitudinal setting. In the proposed model, we consider subject-specific correlation arising from data obtained repeatedly on the same individuals. We utilize an approach similar to the standard SiZer in which a family of smooth functions is used at various levels of smoothing parameters $\lambda$. We enhance the underlying model through the use of a computationally efficient smoothing model (presented below). In the PS-SiZer map, we also apply simultaneous confidence limits to resolve the issues related to multiple comparisons. Our proposed methodology extends SiZer in the following areas:

1. Adding a random intercept component to summarize subject-specific correlation

2. Applying a P-spline [13] as the underlying smoothing function

3. Constructing a simultaneous 95% confidence limit addressing multiple-comparison issues

## Model specification

Let $y_{ij}$ denote measurements on subject $i = 1, 2, \ldots n$ at time $x_{ij}, j = 1,2 \ldots n_i$. We model the responses as,

$$y_{ij} = g_\lambda(x_{ij}) + b_i + \varepsilon_{ij}; \quad \varepsilon_{ij} \sim N(0, \sigma_\varepsilon^2) \tag{2}$$

where $g_\lambda(x_{ij})$ is a smooth function indexed by a smoothing parameter $\lambda$ and $\varepsilon_{ij}$ is a vector of random normal error terms with mean 0 and variance $\sigma_\epsilon^2$. The model in (2) extends the basic model in (1) by adding the random subject-specific component $b_i \sim N\left(0, \sigma_b^2\right)$, a normally distributed random intercept with mean 0 and variance $\sigma_b^2$ which accounts for the within-subject correlation in the repeatedly collected measurements $y_{ij}$ in subject $i$. As this model is a member of the family of linear mixed models, a major advantage from its use is the ability to handle longitudinal data at irregularly spaced time points [14]. In this paper we use the P-spline model of Eilers & Marx [13], as the underlying smoothing method to estimate the function $g_\lambda$. The P-spline model specification includes B-splines as the bases functions with evenly spaced knots with the difference penalty applied directly to the B-spline regression parameters to control the smoothness of the function $g_\lambda$. Let $B_m(x_{ij};p)$ denote B-spline basis of degree $p$ with $k'$ + 1 internal knots. The number of B-splines is $M = k' + 1 + p$ in the regression, resulting in the following approximation of the smooth function $g_\lambda$:

$$g_\lambda\left(x_{ij}\right) = \sum_{m=1}^{M} a_m B_m(x_{ij}; p)$$

where $a_m$ is a vector of coefficients, and $B_m(x_{ij};p)$ is the B-spline basis function of degree $p$. For the penalty term, the P-spline model of Eilers and Marx uses a base penalty on higher-order finite differences, $\Delta_d{}^T\Delta_d$[13]. Consequently, the difference penalty matrix with order $d$ can be written as, $a^T\Delta_d{}^T\Delta_d\, a$. Here, $\Delta_d$ is a matrix such that $\Delta_d$ constructs the vector of $d^{th}$ difference of the coefficients $a$ i.e., $\Delta a_m = a_m - a_{m-1}$; $\Delta^2 a_m - 2a_{m-1} + a_{m-2}$ and so on.

The second component of the model is the addition of a subject-specific random effect $b_i \sim N\left(0, \sigma_b^2\right)$. This results in the penalized least square objective function minimizing

$$\|y - Ba - Zb\|^2 + \lambda a^T\Delta_d{}^T\Delta_d a + (\sigma_\varepsilon^2/\sigma_b^2)b^T b \tag{3}$$

where $Z = \begin{pmatrix} 1_1 & \cdots & 0 \\ \vdots & \ddots & \vdots \\ 0 & \cdots & 1_n \end{pmatrix}$ and $1_i = \begin{bmatrix} 1 \\ \vdots \\ 1 \end{bmatrix}_{n_i \times 1}$

## Mixed model representation

The minimization problem discussed in the previous section can be handled using the mixed-model framework [15]. Eq (3) can be turned into a regular mixed model by making use of the mixed-effect model framework discussed in detail in [1, 16–18] among others. Let us first consider the difference matrix, $\Delta_d$ that has dimension $(k' + 1 + p) \times (k' + 1 + p - d)$. The penalty matrix $\Delta_d^T\Delta_d$ is singular and has rank $(k' + 1 + p - d)$. A singular value decomposition of $\Delta_d^T\Delta_d$ leads to $\Delta_d^T\Delta_d = U diag(\Lambda)U^T$ with $U$ are the eigenvectors and $\Lambda$ is the diagonal matrix of eigenvalues in non-increasing order. Thus, $k' + 1 + p - d$ eigenvalues are strictly positive and the remaining $d$ are zeros. Hence, $U$ and $\Lambda$ can be represented as $U = [U_+, U_0]$ and $\Lambda = (\Lambda_+^T, 0_+^T)^T$ respectively. The dimension of $U_+$ is now $(k' + 1 + p) \times (k' + 1 + p - d)$ with corresponding non-zero elements of vector $\Lambda$. Consequently, we can rewrite $Ba$ as

$$Ba = BUU^Ta = B\left[U_0 U_0^T a + U_+ diag\left(\Lambda_+^{-\frac{1}{2}}\right) diag\left(\Lambda_+^{\frac{1}{2}}\right) U_+^T a\right]$$

$$=: B\left[U_0\beta + U_+ diag\left(\Lambda_+^{-\frac{1}{2}}\right)u\right] =: \mathrm{X}\beta + Z_B u$$

and

$$a^T\Delta_d^T\Delta_d a = a^T U diag(\Lambda) U^T a = a^T U_0 diag\left(0_q\right) U_0^T a + a^T U_+ diag(\Lambda_+) U_+^T a = u^T u$$

The mixed-model representation of the smooth function is $X\beta + Z_B u$, where $u \sim N\left(0, \sigma_u^2 I_{k+1+p-d}\right)$. Our final model, including the random intercept, is of the form,

$$Y = X\beta + Z_B u + b_i + \varepsilon \tag{4}$$

where $u \sim N\left(0, \sigma_u^2 I_{k+1+p-d}\right), b_i \sim N\left(0, \sigma_b^2\right)$ and $\varepsilon \sim N\left(0, \sigma_\varepsilon^2 I_n\right)$.

The model in (4) above has three components. $X\beta$ represents the fixed overall effect while $Z_B u$ corresponds to the smoothing function and $b_i$, subject-specific random intercept, measures the random departure of subject $i$ from the overall effect. The estimates of the parameters and the random coefficients are obtained as the best linear unbiased predictors (BLUP) from the mixed model using the restricted maximum likelihood (REML) criterion for the variance components. Eq (4) can thus be solved using any standard mixed-model software. We utilized the R-package `mgcv::gam` [19], which provides a computationally feasible approach to the

parameter estimation in Eq (4). We obtained the estimate of $g_\lambda(x)$, the mean population curve at $x$, and the quantities of interest to generate the PS-SiZer map as the most crucial component in the PS-SiZer map is to estimate the first derivatives of the fitted functions $\hat{g}_\lambda(x)$(i.e. $\hat{g}_\lambda'(x)$) and the variance of $\hat{g}_\lambda'(x)$ and associated confidence bands. The gam() function from the R library mgcv was used to estimate the function g(x) at varying levels of the smoothing parameter $\lambda$. The PS-SiZer map includes a number of levels of the smoothing parameters. In the present application, we used a range between $log10(\lambda_{REML}) \pm 2$, where $\lambda_{REML}$ is the estimated smoothing parameter obtained via the REML approach using mixed model representation of the P-spline model. At each smoothing level, the resultant smoothing function component is obtained and extracted from the subsequent model fitting.

## Inference

In the previous sections the point estimate for the model parameters were discussed, yet we are also interested in finding the confidence intervals for the quantities derived from them, such as an estimate of smooth function, $\hat{g}_\lambda(x)$ and the first derivatives of the smooth function, $\hat{g}_\lambda'(x)$. We describe the estimate of the covariance matrix for the smoothing parameters specified by [18]. Let $\Phi = \begin{bmatrix} \beta \\ b \end{bmatrix}$ contain all the fixed and random effects from the smooth term only and let $C = \begin{bmatrix} X & Z_B \end{bmatrix}$ be the corresponding model matrix. Let $Z$ be the random effect model matrix excluding the columns corresponding to the smooth terms and $\sigma_b^2$ be the corresponding random effect covariance. The covariance matrix is $V = Z\sigma_b^2 Z^T + \sigma_\epsilon^2 I$. Therefore, the estimated covariance matrix $(\Sigma)$ for the parameters is

$$\Sigma = cov(\Phi) = \left(C^T V^{-1} C + \check{D}\right)^{-1}$$

where $\check{D}$ is the positive semi-definite matrix of the coefficients for the smooth terms. The standard error of the smooth function estimate, $\hat{se}(\hat{g}_\lambda(x)) = \sqrt{C_x(\Sigma)C_x^T}$ with $C_x = [X_x \quad Z_{Bx}]$.

## Estimate and variability bands of the derivatives

The derivatives of the smooth function $g_\lambda(x)$ are obtained by defining $C_x' = \begin{bmatrix} X_x' & Z_{Bx}' \end{bmatrix}$. Here, $X_x' = \frac{d}{dx}(X)$ and $Z_{Bx}' = \frac{d}{dx}(Z_x)$. The first derivative estimate of $\hat{g}_\lambda(x)$ is:

$$\hat{g}_\lambda'(x) = C_x'\Phi$$

The estimated standard error of $\hat{g}_\lambda'(x)$ is $\hat{se}(\hat{g}_\lambda'(x)) \cong \sqrt{C_x'(\sum)\acute{C}_x^T}$.

## Confidence bands

The construction of the PS-SiZer map involves a family of smooth functions based on the confidence bands of the derivatives $\hat{g}_\lambda'(x)$. We used 100 values of the smoothing parameter $\lambda$ on the logarithmic grid to construct the PS-SiZer map. The range of the smoothing parameters $\lambda_{min}, \lambda_{max}$ was chosen as,

$$(\log(\lambda_{\min}), \log S(\lambda_{\max})) = [(\log_{10}(\lambda_{REML}) - 2, \log_{10}(\lambda_{REML}) + 2]$$

The number 2 above is arbitrary. However, the range of the smoothing parameters shown above spans 4 orders of magnitude giving us a picture of the estimated function at a sufficient smoothing span. We obtained the $\lambda_{REML}$ from the REML estimate of the variance components using the same P-spline methodology.

The PS-SiZer can be viewed as a collective summary of a large number of hypothesis tests so multiple-testing issues must be addressed. We follow Ruppert et al., [1] who showed that

the penalized spline has fairly straightforward simulation-based simultaneous confidence bands which can be used in situations when multiplicity testing is carried out. Suppose we want a simultaneous confidence band for $g_\lambda(\cdot)$ on a grid of x-values, $x_{grid} = (x_1, \cdots, x_r)$ and define

$$g_\lambda(x_{grid}) = \begin{bmatrix} g_\lambda(x_1) \\ \vdots \\ g_\lambda(x_r) \end{bmatrix}$$

A $100(1 - \alpha)\%$ simultaneous confidence band for $g_\lambda(x_{grid})$ is

$$\hat{g}_\lambda\left(x_{grid}\right) \pm q_\lambda(1 - \alpha) \begin{bmatrix} \widehat{SD} \ \{\hat{g}_\lambda(x_1) - g_\lambda(x_1)\} \\ \vdots \\ \widehat{SD} \ \{\hat{g}_\lambda(x_r) - g_\lambda(x_r)\} \end{bmatrix}$$

where $\hat{g}_\lambda(x_{grid})$ is the corresponding empirical best linear unbiased predictor (EBLUP) obtained from the mixed model framework. Here, $q_\lambda(1 - \alpha)$ is the $(1 - \alpha)$ quantile of the random variable at a smoothing level $\lambda$, i.e.,

$$\sup_{x \in \mathcal{X}} \left| \frac{\hat{g}_\lambda(x) - g_\lambda(x)}{\widehat{SD} \ \{\hat{g}_\lambda(x) - g_\lambda(x)\}} \right| \tag{5}$$

which is the supremum on the set $\{g_\lambda(x_{grid}) : x \in \mathcal{X}\}$. The quantile $q_\lambda(1 - \alpha)$ was approximated using $N = 10,000$ simulations. The $N$ simulated values were sorted from smallest to largest, and the one with rank $(1 - \alpha)N$ was used as $q_\lambda(1 - \alpha)$. For a PS-SiZer map, we obtained the 95% quantile of Eq (4) based on a simulation of size $N$ at each level of $\lambda$. The confidence limits for $\hat{g}'_\lambda(x)$ were obtained as follows:

$$\hat{g}'_\lambda(x) \pm q_\lambda(1 - \alpha) * \ \widehat{sd} \ (\hat{g}'_\lambda(x))\} \tag{6}$$

## Results

### Simulation studies

In practice, the fundamental function of a SiZer map is to detect the underlying features in the data. For this reason, it is natural to compare the SiZer maps according to which ones detect the correct number of underlying features.

We have conducted Monte Carlo simulation studies to evaluate the performance of PS-SiZer map under various scenarios. The key objective of this simulation study was to compare the PS-SiZer with the LL-SiZer and SS-SiZer. Our simulation studies were designed to mimic the HIV data analyzed later as part of the illustration of the methodology. In doing so, we use the concept of *effective degrees of freedom* (EDF) to encapsulate the complexity of the model as the actual degrees of freedom are not defined for semiparametric models. We use the established method for estimation of the EDF, i.e., the trace of the smoother matrix (see Hastie and Tibshirani [20] as cited in Chauduri and Maron[5]).

Using these simulated data, the relative performance of the SiZer maps was evaluated by the following approaches:

1. By making the SiZer maps comparable at a similar level of Effective Degrees of Freedom. For this reason all three SiZer maps (PS-SiZer with LL-SiZer and SS-SiZer) were generated with the same range of EDF.

2. By comparing the performance of the three SiZer maps at various levels of EDF according to which flags more features of a curve when a curve changes its status (increasing, decreasing or stable) from one to another.

3. By comparing the performance of the three SiZer maps which are most sensitive to detect plateaus that are really present in the data.

In this research, performance of the PS-SiZer maps are presented in two different simulation studies: "Simulation Study 1" which addresses item 2 above and "Simulation Study 2" which addresses item 3.

Simulation study 1. Longitudinal data were simulated as $x_i$ chosen to be equally spaced in the interval [0, 1] with

$$f(x_{ij}) = 65 + 25e^{-2.0*x_{ij}}*\sin(5\pi(x_{ij} + 5)) + b_i + \varepsilon_{ij}$$

where $\varepsilon_{ij} \tilde{} N(0, \sigma_\varepsilon^2)$ is random noise, $b_i \tilde{} N(0, \sigma_b^2)$ is the subject-specific random intercept and $x_{ij}$ denote the time of measurement. The function $\sin(5\pi(x + 5))$ is a periodic function which has five features. By this term, we mean changes in the curve from increasing to decreasing or vice versa. The quantity, $25e^{-2.0x}$ is a function to control the spread of the periodic sine function, which has the effect of diminishing the size of the features at higher time intervals.

We compared three SiZer maps through the various levels of combination of error variance and subject-specific variance, $(\sigma_\varepsilon^2 : \sigma_b^2) = (2 : 5), (5 : 2),$ and $(5 : 15)$ respectively (Table 1). For each scenario of different variance combination, 50 trials were generated consisting of N = 100 subjects each and the number of observations per subject was $n_i = 10$ for $i = 1, \ldots N$. For each simulated trial, three different SiZer maps were generated at 100 levels of EDF.

Table 1 represents the mean proportion of features detected by the 50 simulated data sets at various levels of $\sigma_\varepsilon^2$ and $\sigma_b^2$. All three maps detected the first three features in the data most of the time. We were mainly interested to find how sensitive PS-SiZer is to detect the fourth and the fifth features of the true curve compared to LL-SiZer and SS-SiZer, as these were significantly diminished by the addition of the phasing-out component in the data as described above. When subject-specific variation is small (i.e., $\sigma_b^2 = 2$), PS-SiZer detected all five features 51% of the time, whereas the SS-SiZer and LL-SiZer were able to detect all features 2% and 14% of the time respectively. At the same variability level, four features were detected by

**Table 1. Finding features: Simulation study-1 with varying variability.**

| Variability ($\sigma_\varepsilon^2 : \sigma_b^2$) | Number of features detected | SiZer Maps | | |
|---|---|---|---|---|
| | | LL-SiZer | SS-SiZer | PS-SiZer |
| 5.0: 2.0 | Five | 14% | 2% | 51% |
| | Four | 34% | 47% | 88% |
| 2.0: 5.0 | Five | 4% | 10% | 30% |
| | Four | 32% | 62% | 85% |
| 5.0: 15.0 | Five | 0% | 5% | 8% |
| | Four | 24% | 40% | 68% |

Proportions are from 50 simulation data sets.

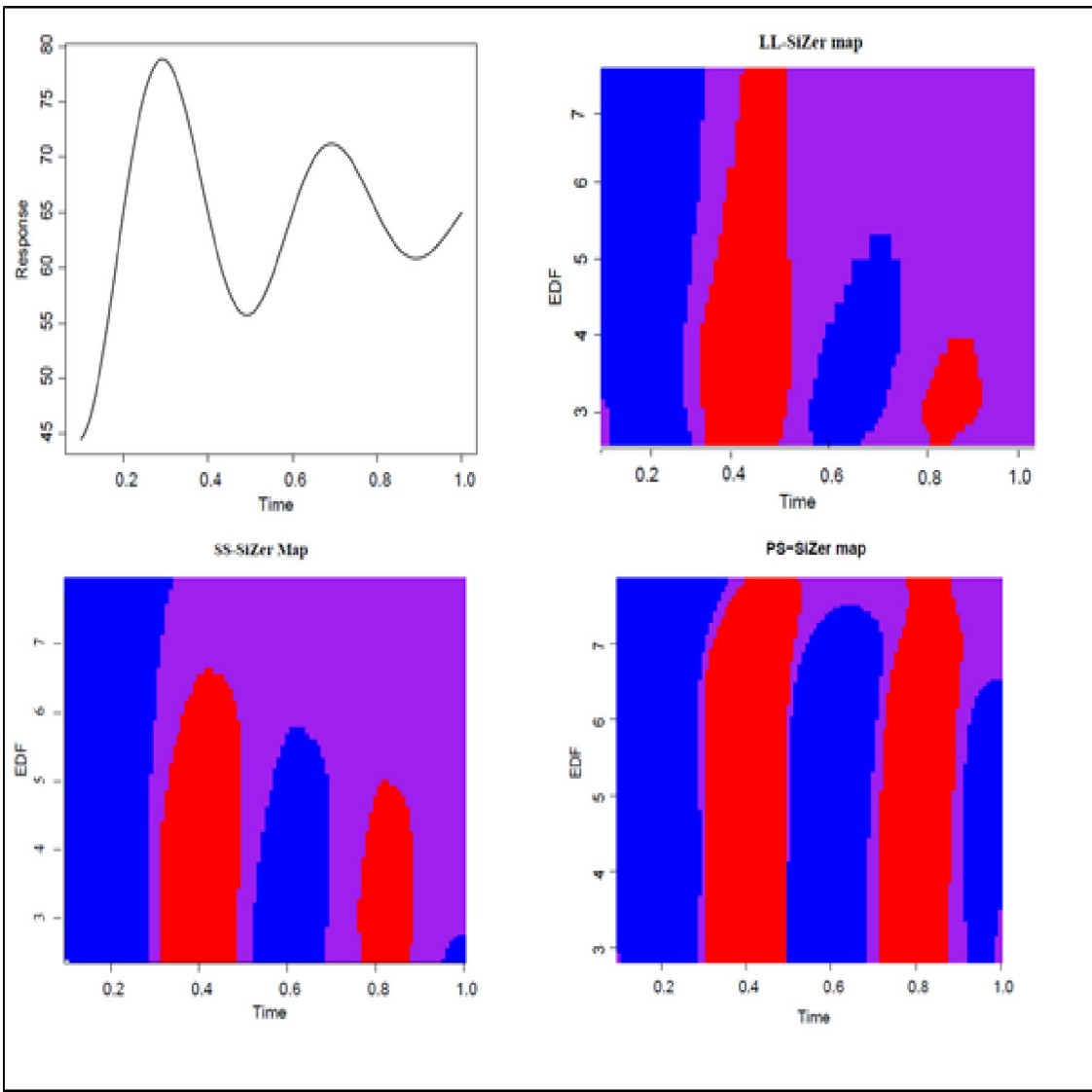

**Fig 2. Simulation study 1.** Upper-left panel: True function; Upper-right panel: LL-SiZer map. Lower-left panel: SS-SiZer map. Lower-right panel: PS-SiZer map. For the SiZer maps, vertical axis represents 100 levels of EDF and the horizontal axis represents time.

PS-SiZer 88% of the time, compared to 47% and 34% by SS-SiZer and LL-SiZer respectively. When the subject-specific variation is high (i.e., $\sigma_b^2 = 15$), PS-SiZer was still able to detect four features in the data almost 68% of the time compared to 40% and 24% by SS-SiZer and LL-Si-Zer respectively. Interestingly, the fifth feature was not detected by LL-SiZer at all in this variability level, compared to 8% by PS-SiZer and 5% by SS-SiZer (Table 1).

Results from the above table are illustrated in Fig 2. Three maps were generated for each of the three methods under comparison from a randomly chosen trial from out of the 50 trials generated in the simulation study $\left(\sigma_\varepsilon^2 : \sigma_b^2\right) = (5 : 15)$. In the Figure, the *x*-axis is represents time and the *y*-axis the EDF or the scale of smoothing of the three maps. As it is clear from the Figure, all three maps were able to clearly flag the dominant first and second features (blue and then red regions in the maps). However, in the majority of smoothing levels, LL-SiZer could

not flag the third or fourth features as being statistically significant and did not detect the fifth feature at any level of EDF as mentioned above in the description of Table 1. SS-SiZer detected all four features for a large proportion of smoothing parameters and was able to detect the fifth feature only at higher levels of EDF, i.e. undersmoothing. By contrast, PS-SiZer detected all five features at the majority of smoothing parameter levels (Fig 2).

**Simulation study 2.** In this simulation study, our aim was to illustrate how sensitive PS-SiZer map is compared to LL-SiZer and SS-SiZer in detecting the plateau of an increasing function. The true curve and the first derivative of the curve are presented in the top left panel of Fig 3. The data were generated as $x_i$ equally spaced in [1:20] with

$$f\left(x_{ij}\right) = 85 - \frac{x_{ij}}{4} - e^{\left(-x_{ij}+4.5\right)} + b_i + \varepsilon_{ij}$$

where $x_{ij}$ are time measurements as before, $\varepsilon_{ij} \sim N\left(0, \sigma_\varepsilon^2 I\right)$ is independent random noise and $b_i \sim N\left(0, \sigma_b^2\right)$ is the subject-specific random intercept. In a manner similar to Simulation Study 1, we generated 50 simulated trials, each with $N = 100$ subjects and involving $n_i = 10$ equally spaced time points $i = 1, \ldots, N$. We consider an error variance $\sigma_\varepsilon^2 = 10$ and a subject-specific random variation $\sigma_b^2 = 5$. For each simulated trial, three SiZer maps were generated at 100 levels of EDFs.

The function used in this example had a true plateau at time $x = 4.5 + \ln\left(\frac{1}{4}\right) \sim 5.89$. The sensitivity of the SiZer maps was calculated at each level of EDF by following exploring at which point in all three SiZer maps, a blue region changed to a purple region at each level of EDF. The process was repeated for 50 simulation trials. The summary of the first time point where the plateau was detected by the three SiZer maps is presented as a box plot (Fig 4). The box plot summary shows that the PS-SiZer map detects the earliest time point of the plateau at $x \sim 5.89$ the closest estimate of the true value. By contrast, LL-SiZer and SS-SiZer detected the plateau of the curve at $x > 6$. The true data curve and the resulting three SiZer maps from a randomly selected simulated trial are presented in Fig 3. Three maps were able to plot the pattern of the curve by moving from the blue region to the purple region at all levels of EDF.

Combined, Simulation Studies 1 and 2 demonstrate that the PS-SiZer map not only detects the significant changes of the true curve, but is also sensitive enough to detect the true time point where the curve reaches its plateau. Even though all three SiZer maps were able to detect the dominant features of the underlying curves, (that is, the trajectory of the curve from significantly increasing–blue area–to decreasing–red area–to non-significant–purple area), they were not able to detect less pronounced features at almost all levels of the EDF and were less sensitive than PS-SiZer in locating the true plateau of the curve.

## Illustration

As an illustration of the proposed methodology of the PS-SiZer, we analyze data on weight changes in people living with HIV who initiate ART. In addition to detecting features in the data corresponding to body weight increases after the start of therapy, an important clinical question pertains to the durability of weight gains under different treatment regimens. More specifically, we explore possible differences in the durability of weight gain between stavudine (d4T) containing ART regimens versus non-d4T-containing regimens. Previous literature suggests that d4T is associated with lipodystrohy, a problem with the way the body produces and stores fat [21] and long-term weight loss compared to other regimens such as, for example, those containing Tenofovir [22] a regimen which is increasingly used as a first-line antiretroviral drug, particularly in the Southern Africa region.

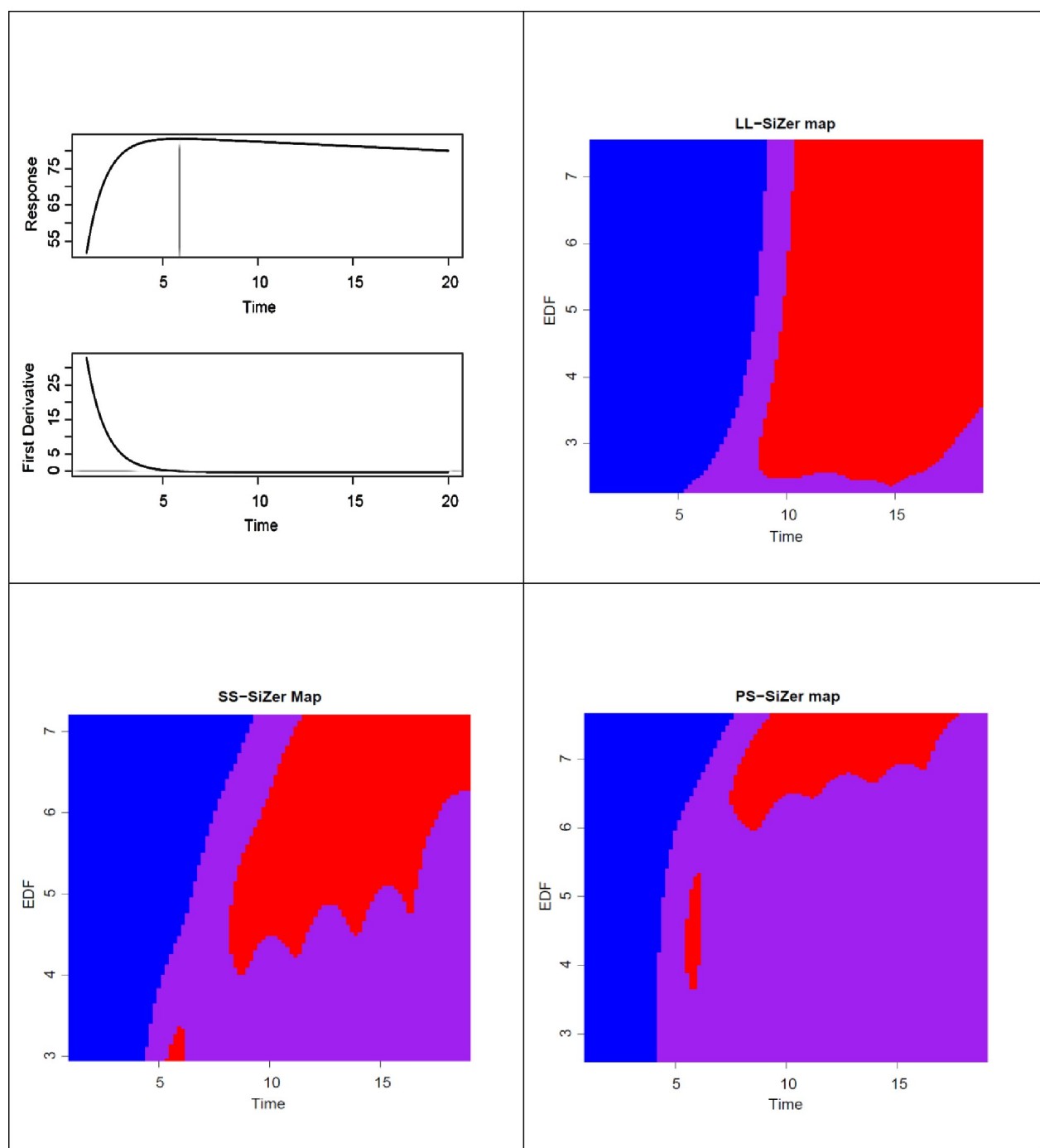

**Fig 3. Simulation study 2.** Upper-left panel: True function and first derivative; Upper-right panel: LL-SiZer map; Lower-left panel: SS-SiZer map; Lower-right panel: PS-SiZer map. The vertical axis represents the 100 levels of EDF and the horizontal axis represents the time.

The present study includes data on 185,010 adults living with HIV from five regions within the IeDEA collaboration [23]: Southern Africa (65.6% of the cohort), East Africa (21.9%), West Africa (8.3%), Central Africa (3.2%) and Asia Pacific (0.9%). Baseline demographic data of IeDEA patients identified by region and by d4t-containing versus non-d4T-containing regimen are shown in Table 2. In the sequel, we present in detail results from the Southern Africa

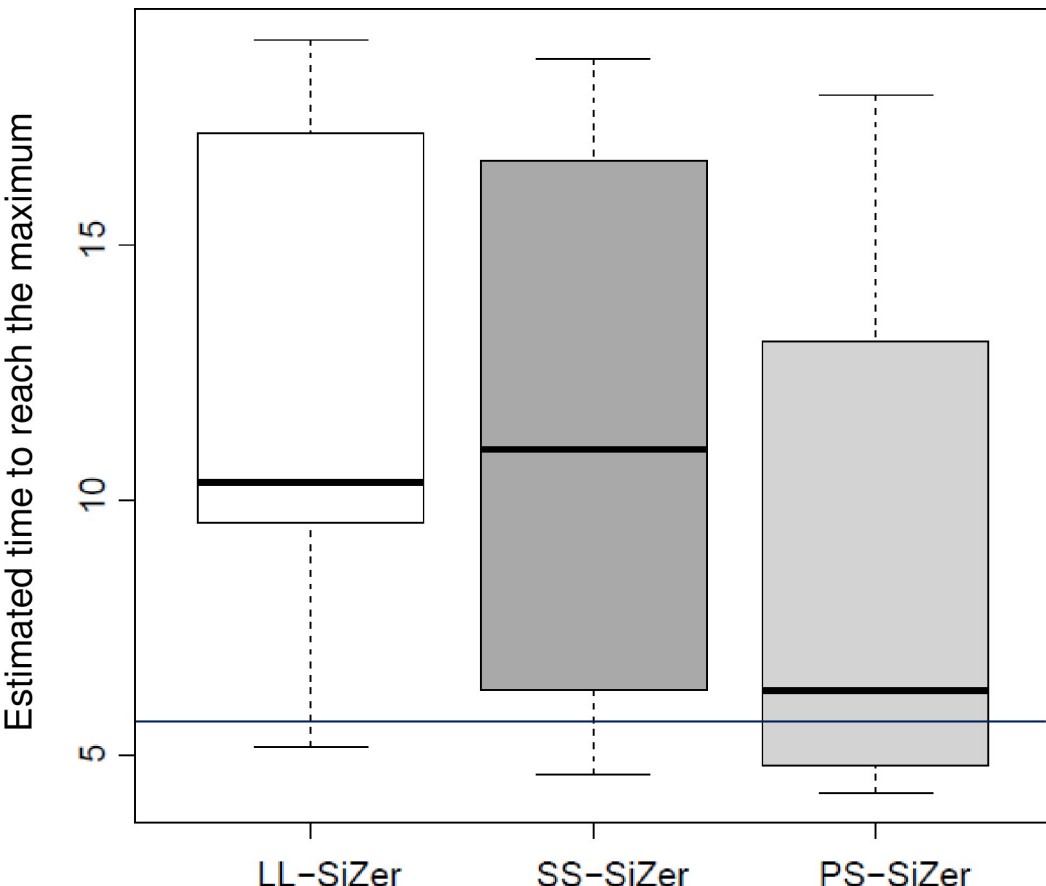

**Fig 4. Boxplot-summary of three SiZer maps: Time to detect a true plateau in the data.**

IeDEA region. Results from the remaining four regions are presented in less detail and are left for the supplementary material.

PS-SiZer Maps for the Southern IeDEA region were generated for each ART group, i.e., one map each for the groups of patients initiating ART with a regimen containing or not containing d4T. To address the issue of durability of weight gain, we need to determine the first time point (in weeks from start of ART) at which weight gain stops, i.e., the time when either weight stops increasing or starts to decrease. The PS-SiZer map provides an overall visual

**Table 2. Summary of baseline characteristics-IeDEA study by d4T and non-d4T based regimen.**

| | d4T Regimen | | | | Non-d4T regimen | | | |
|---|---|---|---|---|---|---|---|---|
| | N | Age (years) | Female (%) | Baseline Body weight (kg) | N | Age (years) | Female (%) | Baseline Body weight (kg) |
| **Overall** | 98160 | 36(30–42) | 64152 (65) | 55.0 (48–62) | 86850 | 36 (30–42) | 50682 (58) | 55.0(49–62) |
| **Asia Pacific** | 963 | 35(29–40) | 410 (43) | 51.0 (45–58) | 751 | 34 (29–42) | 181 (24) | 57.7(50–56) |
| **Central Africa** | 2839 | 37 (31–44) | 2008 (70) | 56.0 (49–65) | 3045 | 37 (31–44) | 2118 (51) | 56.0 (50–65) |
| **East Africa** | 30990 | 37 (31–43) | 20017 (78) | 54.0 (48–61) | 9571 | 37 (31–43) | 5758 (22) | 55.0 (49–62) |
| **Southern Africa** | 55192 | 35 (30–42) | 36227 (49) | 55.0 (48–62) | 66295 | 35 (30–42) | 38137 (51) | 55.0 (49–62) |
| **West Africa** | 8176 | 39 (32–42) | 5490 (55) | 55.0 (48–64) | 7188 | 41 (37–42) | 4488 (45) | 57.0 (50–65) |

Summaries are median (IQR) or n (%)

representation of the longitudinal weight change after the start of ART. However, to reach a conclusion on the durability of weight increases after ART start, we need to decide on a single optimum level of smoothing. Our algorithm does not depend on a specific smoothing technique. Here we have used a P-spline (13) PSR model for its computational efficiency and flexibility for correlated data. In addition, we took advantage of re-expressing the PSR model as a linear mixed effect model (1). The REML estimate of the mixed model is used to obtain the optimum smoothing parameter.

Statistical analyses were performed using SAS Software 9.3 and R software (2.13.2). SAS was used to create the analysis datasets for each of the five IeDEA regions. The user defined R-functions and the R package `SiZer` [24] was used to generate LL-SiZer maps. To generate SS-SiZer and PS-SiZer maps, user defined R-functions and the R package `mgcv::gam` [19] was used.

The PS-SiZer map for the Southern-Africa region is presented in Fig 5. The corresponding maps generated for each of the remaining four IeDEA regions are presented in S1–S4 Figs. Each Figure is divided into four panels. The smoothed trajectories of weight after ART initiation at the optimum level of smoothing for the two regimens and the smoothed first derivative of the weight change over time is the two types of regimens are shown in the top row. The PS-SiZer maps for d4T-containing and non-d4T-containing ART regimens are shown in the top row. In each PS-SiZer map, the vertical axis represents the level of smoothing and the horizontal axis the time in weeks since the start of ART as described in the Methods. For example, for d4T-containing regimens, at a medium level of smoothing (0.5–1.0), body weight increases for about 60 weeks, as reflected by the blue color on the left of the PS-SiZer map. The area to the right of the blue region is colored purple, indicating that no more significant increases in body weight are evident after about 60 weeks from the start of ART. There are red regions in the map at most of the smoothing levels indicating possible weight decreases. Similarly, at very high smoothing levels, (i.e., for values of the smoothing parameter $\lambda > 1.0$), the entire map is blue, following purple and red indicating weight increases, then stable or decreasing for the entire follow-up period. The PS-SiZer map of body-weight changes among individuals initiating ART with a non-d4T-containing regimen shows that, at lower smoothing levels, there are some blue and purple areas suggesting an intermittent weight increase. Otherwise, the map consists of mostly blue areas (indicating weight increases) for medium and higher levels of smoothing for up to about 150 weeks after ART initiation. This indicates that patients starting ART with a non-d4T-containing regimen experience sustained body-weight increases for a period possibly double that of patients treated with d4T-containing regimens.

To reach a conclusion about the comparison of the durability of weight changes in the d4T-containing versus not-containing ART regimens, we choose the PS-SiZer analysis (top-row of the PS-SiZer map) at the optimum level of smoothing for the Southern Africa IeDEA region (Fig 5). This analysis shows that weight in patients treated with d4T-containing ART regimens increased rapidly after ART initiation and plateaued afterwards. Consulting the first derivative (Fig 5 top-row right panel), we observed that the 95% CI of the curve includes zero after 59.9 weeks in the group of patients who received a d4T-containing regimen compared to 133.8 weeks for patients treated with non-d4T-containing regimens. A numerical summary of these results is shown in the first row of Table 3.

Similar analyses are presented in S1–S4 Figs for the East-Africa, West Africa, Central-Africa, and Asia-Pacific IeDEA regions respectively. The SiZer maps corresponding to the East and West Africa regions are very similar. For d4T-containing regimens (panel a1 in S1 and S2 Figs), blue areas are followed by purple areas after about 50 weeks for most levels of smoothing, indicating significantly increasing weight during this period. After this point, weight gain diminishes. By contrast, the blue areas in the SiZer maps corresponding to the non-d4T-

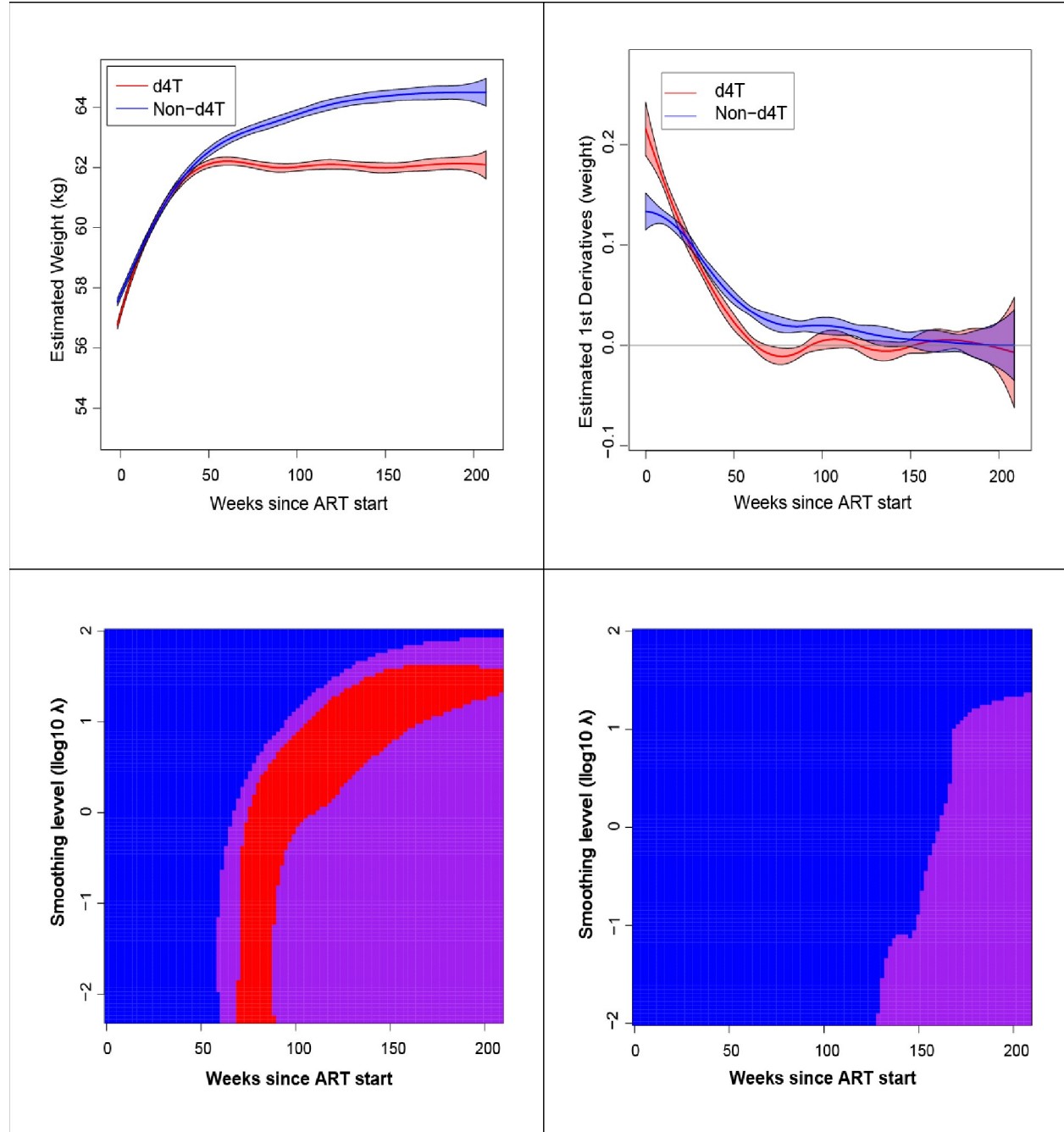

**Fig 5. Southern Africa: Plots of the weight change and its first derivative (top row) and PS-SiZer Maps (bottom row), for d4T-containing and non-d4T-contiainging ART regimens (left and right column respectively).**

containing regimens (S1 and S2 Figs) extend past week 80, indicating that weight continues to increase past 80 weeks after initiation of ART. Analyses at the optimum smoothing level produced the estimated curves of weight measurements are shown in S1 and S2 Figs and rows 2 and 3 in Table 3. For East Africa, results at the optimal smoothing levels showed that the weight in patients treated with d4T-containing regimens did not significantly increase after 52.9 weeks compared to 84.9 weeks for patients treated with non-d4T-containing regimens.

**Table 3. Estimated weeks at which HIV-patients experienced non-increasing weight.**

| | Durability of weight gain | |
| --- | --- | --- |
| | Weeks after ART start (95% confidence interval) | |
| **IeDEA Region** | **d4T-based regimen** | **Non-d4T regimen** |
| Southern Africa | 59.92 (57.56, 62.27) | 133.82 (131.08, 136.56) |
| East Africa | 52.92(50.76, 55.08) | 84.88 (80.57, 89.19) |
| West Africa | 43.94 (39.43, 48.45) | 92.87 (86.59, 99.14) |
| Central Africa | 61.92 (54.86, 68.98) | 60.92 (53.23, 68.37) |
| Asia-Pacific | 38.94 (34.45, 43.43) | 54.92 (46.69, 63.15) |

For West Africa, the results are similar, with d4T-containing regimens estimated to weight gain for 43.9 weeks versus 92.9 weeks for the non-d4T-containing regimens.

Results were similar in analyses from data in the Central-Africa and Asia-Pacific IeDEA regions (S3 and S4 Figs respectively) but the differences between the two regimens were less pronounced. Analyses of data from the Central Africa IeDEA region are shown in S3 Fig and in row 4 of Table 3. The estimated duration of weight increases in the Central Africa region was 61.9 weeks for d4T-containing ART regimens versus 60.9 weeks for non-d4T-containing regimens.

Results from the analyses of data in the Asia Pacific IeDEA region are presented in Fig 3 and row 5 of Table 3 The estimated duration of weight gain in d4T-containing regimens was 38.9 weeks versus 54.9 weeks in non-d4T-containign regimens.

## Discussion

This paper presents a significant extension of the SiZer methodology, the penalized SiZer or PS-SiZer. Current SiZer methods, such as the standard LL-SiZer [5] and SS-SiZer [3] do not account for the correlation induced by repeated measurements obtained on the same patient, which invariably arise in longitudinal settings with particular frequency in biomarker data. In addition, to developing a SiZer variant which can take into account within-subject correlation, our efforts were also centered on developing computationally efficient methods to address analyses involving massive databases from tens of thousands of subjects and millions of individual measurements.

The fundamental motivation of the originally proposed SiZer map is to detect the underlying features in the data and present a global visualization of changes in quantitative data for a spectrum of smoothing levels. The key goal of this research is to show propose a SiZer variant which can detect more real features in data in the context of data collected repeatedly from the same subjects at irregular time points longitudinally. From the simulation results, it was evident that both the standard LL-SiZer formulation and the SS-SiZer method, while able to detect large dominant features in the data, missed more subtle features, because neither method appropriately addresses within-subject variability. This results in wider confidence intervals and a diminished sensitivity when features in the data become attenuated (i.e., smaller changes from increases to decreases or vice versa).

Marron & Zhang [3] have also attempted to compare these two maps by carrying out a number of simulations studies. The authors concluded that the original local linear version (here, LL-SiZer) of the SiZer and the smoothing spline SiZer (here, SS-SiZer) often performed similarly, without one method dominating the other in all cases. Similar findings were observed in our own simulation studies. By contrast, the PS-SiZer maps identified more underlying features in the simulation data than the other two SiZer map methods at a wide

range of smoothing levels. In addition, both LL-SiZer and SS-SiZer detected a plateau in the data later compared to the PS-SiZer map, which detected the plateau almost exactly at the true time that it occurred in the simulated data. The simulation studies thus clearly demonstrate that, at a wide range of smoothing levels, PS-SiZer was more sensitive to small features in the data than the other two methods, presumably due to its improved ability to account for the presence of correlation in the data. More recently, Chen & Wang[25] proposed a new method for using the penalized spline approach for functional mixed effects models with varying coefficients. Their focus is different from our approach, which is used for the discovery of features in the underlying population regression function, by expanding the applicability of the SiZer approach to longitudinal designs where the P-spline model of Eilers & Marx [13] is used to estimate the population regression curve.

The main idea of SiZer maps is to detect significant changes in the data by mapping areas where the 95% confidence intervals of the first derivative is significantly different from zero. The combination of the penalized spline regression model with random intercepts in the PS-SiZer map results in narrower confidence intervals, which, in turn, lead to more sensitive detection of even less prominent features present in the data compared to standard SiZer maps. In summary, PS-SiZer is a reasonably accurate addition to the family of SiZer map methods particularly when analyzing data from longitudinal settings.

In the application of the PS-SiZer methodology, we analyzed a database involving more than 185,000 adult HIV-infected patients and well over two million longitudinal weight measurements. Our ability to handle such a large data set, underlines the computational advantages of the proposed methodology. In addition to a global visualization of the data, the PS-SiZer analysis produced meaningful clinical results by showing that the durability of weight gain experienced by after starting ART with regimens containing d4T is likely significantly shorter than among persons who start ART with regimens which do not contain d4T.

Specifically, within the Southern Africa region, weight increases in the former regimens were observed to end after about 60 weeks from initiating of ART compared to almost 133 weeks (2.5 years) among patients who started ART with regimens not containing d4T. While the clinical importance of this finding is less pronounced, given the almost universal phasing out of stavudine as a first-line regimen, weight gain among people living with HIV is a relevant topic, particularly with the wide adoption of integrase inhibitors and dolutegravir in particular, as main line antiretroviral therapies, all of which are known to result in significant weight gain in these patients [26–28].

These analyses underscore the power of the methodology to detect meaningful features in the data and can address similar questions with other biomarkers, particularly in situations where normalization of the marker is of significant clinical importance. For example, the durability of increases in CD4-positive T-lymphocytes after ART initiation [29] or normalization of inflammatory factors [30] is of major clinical significance in the setting of antiviral treatment of people living with HIV as are numerous other cases of biomarkers, where the timing of normalization of the marker following initiation of therapy can be estimated by the PS-SiZer based on repeatedly obtained measures obtained on the same subjects over time.

## Supporting information

**S1 Fig. East Africa: Plots of the weight change and its first derivative (top row) and PS-Si-Zer maps (bottom row), for d4T-containing and non-d4T-contiainging ART regimens (left and right column respectively).**
(TIF)

**S2 Fig. West Africa: Plots of the weight change and its first derivative (top row) and PS-Si-Zer maps (bottom row), for d4T-containing and non-d4T-contiainging ART regimens (left and right column respectively).**
(TIF)

**S3 Fig. Central Africa: SiZer maps (top row) and plots of the weight change and its first derivative (bottom row), for d4T containing and non-d4T-contiainging ART regimens (left and right column respectively).**
(TIF)

**S4 Fig. Asia Pacific: Plots of the weight change and its first derivative (top row) and PS-Si-Zer maps (bottom row), for d4T-containing and non-d4T-contiainging ART regimens (left and right column respectively).**
(TIF)

## Author Contributions

**Conceptualization:** Jaroslaw Harezlak, Constantin Theodore Yiannoutsos.

**Data curation:** Jaroslaw Harezlak, Samiha Sarwat, Michael Schomaker, Eric Balestre, Matthew Law, Sasisopin Kiertiburanakul, Diana Huis in 't Veld, Beverly Sue Musick, Constantin Theodore Yiannoutsos.

**Formal analysis:** Samiha Sarwat, Constantin Theodore Yiannoutsos.

**Funding acquisition:** Kara Wools-Kaloustian, Matthew Law, Constantin Theodore Yiannoutsos.

**Methodology:** Jaroslaw Harezlak, Samiha Sarwat, Michael Schomaker, Matthew Fox, Diana Huis in 't Veld.

**Resources:** Constantin Theodore Yiannoutsos.

**Supervision:** Jaroslaw Harezlak, Kara Wools-Kaloustian, Constantin Theodore Yiannoutsos.

**Validation:** Beverly Sue Musick.

**Writing – original draft:** Jaroslaw Harezlak, Samiha Sarwat, Constantin Theodore Yiannoutsos.

**Writing – review & editing:** Jaroslaw Harezlak, Kara Wools-Kaloustian, Michael Schomaker, Eric Balestre, Matthew Law, Sasisopin Kiertiburanakul, Matthew Fox, Diana Huis in 't Veld, Beverly Sue Musick, Constantin Theodore Yiannoutsos.

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
