## [Decision Letter · Decision Letter 0]

10 Sep 2019

PONE-D-19-18970

SiZer Map to investigate significant features of body-weight profile changes in HIV infected patients in the IeDEA Collaboration

PLOS ONE

Dear Dr. Yiannoutsos,

Thank you for submitting your manuscript to PLOS ONE. After careful consideration, we feel that it has merit but does not fully meet PLOS ONE’s publication criteria as it currently stands. Therefore, we invite you to submit a revised version of the manuscript that addresses the points raised during the review process.

We would appreciate receiving your revised manuscript by Oct 25 2019 11:59PM. To enhance the reproducibility of your results, we recommend that if applicable you deposit your laboratory protocols in protocols.io, where a protocol can be assigned its own identifier (DOI) such that it can be cited independently in the future. For instructions see: http://journals.plos.org/plosone/s/submission-guidelines#loc-laboratory-protocols

We look forward to receiving your revised manuscript.

Kind regards,

Ram Chandra Bajpai, Ph.D.

Academic Editor

PLOS ONE

Journal Requirements:

2. In your Methods section, please provide additional information about the data set used fro illustration (for example, citing the appropriate references, and describing the data sets in more detail).

3. Our internal editors have looked over your manuscript and determined that it may be within the scope of our Mathematical Modelling of Infectious Disease Dynamics Call for Papers. The Collection will encompass a diverse range of research articles on using mathematical models to better understand infectious diseases. Additional information can be found on our announcement page: https://collections.plos.org/s/mathematical-disease-dynamics. If you would like your manuscript to be considered for this collection, please let us know in your cover letter and we will ensure that your paper is treated as if you were responding to this call. If you would prefer to remove your manuscript from collection consideration, please specify this in the cover letter. " 2) please request the following from the authors and do not ping for follow up: "Please note that PLOS ONE has specific guidelines on software sharing (http://journals.plos.org/plosone/s/materials-and-software-sharing#loc-sharing-software) for manuscripts whose main purpose is the description of a new software or software package. In this case, new software must conform to the Open Source Definition (https://opensource.org/docs/osd) and be deposited in an open software archive. Please see http://journals.plos.org/plosone/s/materials-and-software-sharing#loc-depositing-software for more information on depositing your software.

[Data collection was funded by the NIH National Institute of Allergies and Infectious

Diseases (NIAID). Samiha Sarwat was supported in part by Grant Number TL1 TR000162 (A.

Shekhar, PI) from the National Institutes of Health, National Center for Advancing Translational

Sciences, Clinical and Translational Sciences Award]

 [All authors' institutions received direct or indirect research funding by the National Institutes of Health supporting this study.

The funders had no role in study design, data collection and analysis, decision to publish, or preparation of the manuscript and no funds were directly provided to any of the authors.]

6. Thank you for stating the following in the Competing Interests section:

[The authors have declared that no competing interests exist.].   

* We note that one or more of the authors are employed by a commercial company: 'Bayer U.S., LLC'.

Reviewers' comments:

Reviewer's Responses to Questions

**Comments to the Author**

1. Is the manuscript technically sound, and do the data support the conclusions?

Reviewer #1: Yes

Reviewer #2: Yes

2. Has the statistical analysis been performed appropriately and rigorously? 

Reviewer #1: Yes

Reviewer #2: Yes

3. Have the authors made all data underlying the findings in their manuscript fully available?

Reviewer #1: Yes

Reviewer #2: Yes

4. Is the manuscript presented in an intelligible fashion and written in standard English?

Reviewer #1: Yes

Reviewer #2: Yes

5. Review Comments to the Author

Reviewer #1: In this manuscript, the authors presented an improved method for SiZer, PS-SiZer, which aims at increasing sensitivity and accuracy in detecting relevant features. On simulated data, PS-SiZer has better sensitivity than existing methods. On real data, PS-SiZer can find useful features and biological inferences can be made through these features. The manuscript will be good for publication if the authors solve the following issues:

1. In Table 1, the authors show that PS-SiZer detect a much higher fraction of features than the other two methods. In Figure 2, the authors show that PS-SiZer can detect all features in the simulated study. In Figure 3, the estimated time to reach the plateau is much smaller in PS-SiZer than in other two methods. Taking these results together, is PS-SiZer overly sensitive? If not, can the authors perform another simulation to prove that?

2. The authors need to increase the resolution of figure 1a and 1b, and make figure 1c axis ticks bigger. There are also some typos, e.g. in page 10 "a family of smoonth".

Reviewer #2: Manuscript ID: PONE-D-19-18970

Title: SiZer Map to investigate significant features of body-weight profile changes in HIV infected patients in the IeDEA Collaboration

Summary:

The paper extends the method of SiZer maps to detect the time reaching a plateau when analyzing irregular longitudinal data by using penalized spline regression model (PS-SiZer).

Strengths:

1. It is nice to realize how the penalized spline regression model can be converted into the mixed-model framework to increase the computational efficiency.

2. Methodology development is well described from model specification, inference, estimate of first derivate, to confidence band.

3. Two simulation studies indicate better performance of PS-SiZer in detecting peaks and the time at peaks.

Comments:

1. Please clarify what Effective Degrees of Freedom (EDF) is and the purpose.

2. It is unclear how the model addresses the irregular longitudinal data.

3. The method section indicates the use of R package, mgcv::gam, as mixed model to estimate g(x) function, but data analysis used R package SiZer. Please clarify how g(x) function is estimated.

4. Is any difference of the proposed method vs “A Penalized Spline Approach to Functional Mixed Effects Model Analysis” by Chen and Wang (Biometrics, 2010)?

5. Please illustrate more how the optimum smoothing parameter is estimated using the ‘Rule of thumb’ approach.

6. PLOS authors have the option to publish the peer review history of their article (what does this mean?). If published, this will include your full peer review and any attached files.

Reviewer #1: No

Reviewer #2: No

---

## [Author Response · Author response to Decision Letter 0]

13 Dec 2019

Response to reviewers in attached file.

Regarding any conflict of interest of Dr. Sarwat, the work was completed during her doctoral research at Indiana University and prior to the commencement of her employment with Bayer. She has no conflicts and no commercial interests related to this work and this work is unrelated to her present employment.

As stated in the cover letter, we would like this manuscript to be considered for the special issue of Modeling in Infectious Diseases Call for Papers.

---

## [Decision Letter · Decision Letter 1]

31 Jan 2020

PONE-D-19-18970R1

PS-SiZer Map to investigate significant features of body-weight profile changes in HIV infected patients in the IeDEA Collaboration

PLOS ONE

Dear Dr. Yiannoutsos,

Thank you for submitting your manuscript to PLOS ONE. After careful consideration, we feel that it has merit but does not fully meet PLOS ONE’s publication criteria as it currently stands. Therefore, we invite you to submit a revised version of the manuscript that addresses the points raised during the review process.

Reviewer have highlighted that authors did not incorporate previous comments. Therefore, authors should clearly mention that how they have  incorporated those comments in the manuscript.

We would appreciate receiving your revised manuscript by Mar 16 2020 11:59PM. To enhance the reproducibility of your results, we recommend that if applicable you deposit your laboratory protocols in protocols.io, where a protocol can be assigned its own identifier (DOI) such that it can be cited independently in the future. For instructions see: http://journals.plos.org/plosone/s/submission-guidelines#loc-laboratory-protocols

We look forward to receiving your revised manuscript.

Kind regards,

Ram Chandra Bajpai, Ph.D.

Academic Editor

PLOS ONE

Reviewers' comments:

Reviewer's Responses to Questions

**Comments to the Author**

1. If the authors have adequately addressed your comments raised in a previous round of review and you feel that this manuscript is now acceptable for publication, you may indicate that here to bypass the “Comments to the Author” section, enter your conflict of interest statement in the “Confidential to Editor” section, and submit your "Accept" recommendation.

Reviewer #2: (No Response)

2. Is the manuscript technically sound, and do the data support the conclusions?

Reviewer #2: Yes

3. Has the statistical analysis been performed appropriately and rigorously? 

Reviewer #2: Yes

4. Have the authors made all data underlying the findings in their manuscript fully available?

Reviewer #2: Yes

5. Is the manuscript presented in an intelligible fashion and written in standard English?

Reviewer #2: Yes

6. Review Comments to the Author

Reviewer #2: The responses were not incorporated in the revised manuscript. Here is one example for illustration.

1. Please clarify what Effective Degrees of Freedom (EDF) is and the purpose.

Response: We have clarified that effective degrees of freedom encapsulate the complexity of the model as the actual degrees of freedom are not defined for the semiparametric models. We use the established method for their estimation, i.e. the trace of the smoother matrix (see Hastie & Tibshirani (1990) as cited in Chauduri & Marron (1999, pp. 812)).

The revised manuscript did not have any change to better explain EDF.

By making the SiZer maps comparable at a similar level of “Effective Degrees of Freedom” (EDF). For this reason all three SiZer maps (PS-SiZer with LL-SiZer and SS-SiZer) were generated with the same range of EDFs.

7. PLOS authors have the option to publish the peer review history of their article (what does this mean?). If published, this will include your full peer review and any attached files.

Reviewer #2: No

---

## [Author Response · Author response to Decision Letter 1]

3 Feb 2020

The point about the EDFs is well taken and we apologize for having complicated the review. We have added the text into the body of the revised paper. To facilitate with identifying the changes made in the document, we have added comments in the margins where the precise response and the responded-to comment were identified.

---

## [Decision Letter · Decision Letter 2]

26 Feb 2020

PS-SiZer Map to investigate significant features of body-weight profile changes in HIV infected patients in the IeDEA Collaboration

PONE-D-19-18970R2

Dear Dr. Yiannoutsos,

We are pleased to inform you that your manuscript has been judged scientifically suitable for publication and will be formally accepted for publication once it complies with all outstanding technical requirements.

With kind regards,

Ram Chandra Bajpai, Ph.D.

Academic Editor

PLOS ONE

Additional Editor Comments (optional):

Reviewers' comments:

Reviewer's Responses to Questions

**Comments to the Author**

1. If the authors have adequately addressed your comments raised in a previous round of review and you feel that this manuscript is now acceptable for publication, you may indicate that here to bypass the “Comments to the Author” section, enter your conflict of interest statement in the “Confidential to Editor” section, and submit your "Accept" recommendation.

Reviewer #2: All comments have been addressed

2. Is the manuscript technically sound, and do the data support the conclusions?

Reviewer #2: Yes

3. Has the statistical analysis been performed appropriately and rigorously? 

Reviewer #2: Yes

4. Have the authors made all data underlying the findings in their manuscript fully available?

Reviewer #2: Yes

5. Is the manuscript presented in an intelligible fashion and written in standard English?

Reviewer #2: Yes

6. Review Comments to the Author

Reviewer #2: (No Response)

7. PLOS authors have the option to publish the peer review history of their article (what does this mean?). If published, this will include your full peer review and any attached files.

Reviewer #2: No

---

## [Editor Report · Acceptance letter]

25 Mar 2020

PONE-D-19-18970R2 

PS-SiZer Map to investigate significant features of body-weight profile changes in HIV infected patients in the IeDEA Collaboration 

Dear Dr. Yiannoutsos:

I am pleased to inform you that your manuscript has been deemed suitable for publication in PLOS ONE. Congratulations! Your manuscript is now with our production department. 

With kind regards,

on behalf of

Dr. Ram Chandra Bajpai 

Academic Editor

PLOS ONE